# Collision risk of bats with small wind turbines: Worst-case scenarios near roosts, commuting and hunting structures

Stefanie A. Hartmann[1]*, Klaus Hochradel[2], Sören Greule[3], Felix Günther[3], Bruntje Luedtke[1], Horst Schauer-Weisshahn[1], Robert Brinkmann[1]

1 Freiburg Institute of Applied Animal Ecology, Freiburg, Germany, 2 UMIT—Private University for Health Sciences, Medical Informatics and Technology GmbH, Hall in Tirol, Austria, 3 OekoFor GbR, Freiburg, Germany

* hartmann@frinat.de

**Data Availability Statement:** All relevant data are within the paper and its Supporting Information files.

## Abstract

Small wind turbines (SWTs) have become increasingly common within the last decade, but their impact on wildlife, especially bats, is largely unknown. We conducted an operational experiment by sequentially placing a mobile SWT with five different operational modes at six sites of high bat activity, including roosts, commuting structures, and highly frequented hunting areas. Bat flight trajectories around the SWT were documented at each site during five consecutive nights using a specifically designed high-spatial-resolution 3D camera. The recordings showed high bat activity levels close to the SWT (7,065 flight trajectories within a 10-m radius). The minimum distance to the rotor of each trajectory varied between 0 and 18 m, with a mean of 4.6 m across all sites. Linear mixed models created to account for site differences showed that, compared to a reference pole without a SWT, bats flew 0.4 m closer to the rotor (95% CI 0.3–0.6 m) if it was out of operation and 0.3 m closer (95% CI 0.1–0.4 m) if it was moving slowly. Exploratory behavior was frequently observed, with many bats deviating from their original flight trajectory to approach the rotor. Among 7,850 documented trajectories, 176 crossed the rotor, including 65 while it was in motion. The collision of one *P. pygmaeus* individual occurred during the experiment. These results demonstrate that, despite the generally strong ability of bats to evade moving rotor blades, bat casualties at SWTs placed at sites of high bat activity can reach or exceed the current threshold levels set for large wind turbines. As SWTs provide less energy than large turbines, their negative impact on bats should be minimized by avoidance measures such as a bat-friendly site selection or curtailment algorithms.

## Introduction

Reducing carbon emissions is a major goal within worldwide efforts to mitigate global climate change. As a consequence, the renewable energy sector has rapidly expanded within the last decade, with wind power being among the most widely used forms of alternative energy [1,2].

**Funding:** This project was funded by the German Bundesamt für Naturschutz, FGII 4.3/Naturschutz und erneuerbare Energien, Projekt zum Forschungs- und Entwicklungsvorhaben aus Mitteln des Bundesministeriums für Umwelt, Naturschutz und nukleare Sicherheit (BMU), Einzelplan 16, Kapitel 1604, Titel 54401. Förderkennzeichen: 3517860600 The funder provided support in the form of salaries for authors [S.A.H., S.G, F.G., B.L., H-S.-W, R.B.], but did not have any additional role in the study design, data collection and preparation of the manuscript. The decision to publish internationally was made together with the funder. The additional salary necessary for the preparation of the english manuscript as well as the revisions was covered by FrInaT GmbH. The funders had no role in study design, data collection and analysis, preparation of the manuscript, but read the final manuscript and were part of the decision where to submit the manuscript.

**Competing interests:** During the time of the conduction of the study, all authors except Klaus Hochradel have been affiliated with FrInaT GmbH. The project itself was based with FrInaT. Once the project was finished, two authors (S.G., F.G.) were hired by Oekofor GbR. Oekofor and FrInaT collaborate with several projects and have no competing interests in a financial, professional or personal aspect. The commercial affiliations to FrInaT GmbH and Oekofor GbR do not alter our adherence to PLOS ONE policies on sharing data and materials.

Despite their positive effects on the climate, wind farms can severely impact the local fauna [3], especially bats and birds [4]. Indeed, direct collisions with rotor blades and, to a lesser extent, barotrauma account for a significant proportion of bat deaths [5–7]. The particular vulnerability of bats is in part due to their relatively low reproductive rates and their extended lifespan [8,9]. Population viability studies have shown that the collisions of bats with turbines can lead to plummeting population sizes [10–12] and have stimulated debate on whether priority should be given to climate or to species protection [13].

Although previous studies of bat collisions predominantly focused on large wind turbines, the installation of small wind turbines (SWTs, typically < 15 kW power, < 30 m hub height) has increased, particularly in the private sector, but their effects on wildlife is thus far unknown [14]. Extrapolation of the findings of studies on large wind turbines to SWTs is problematic for at least two reasons: First, bat fatalities are influenced by tower height, rotor size, and revolutions/min, all of which differ significantly between small and large wind turbines [6]. Second, unlike large wind turbines, SWTs are often placed much closer to the homes, farms, or factories of their owners and therefore in closer proximity to hedgerows, gardens, buildings, and other habitat structures known to be attractive to bats. As environmental impact assessments are typically not required for SWTs [14], their damage to bat populations has yet to be assessed.

Preliminary studies have found evidence of negative effects of SWTs on bats, albeit less severe than those of large wind turbines [15–17]. However, those studies addressed only the long-term effects, as all of the included SWTs had already been operational for several years prior to the study period. Since bats tend to increase their exploratory behavior when confronted with new objects but make use of longer echolocation intervals in known vs. unknown areas [18], the installation of a SWT within a formerly undisturbed bat habitat may result in a collision risk that is initially high but then decreases as the SWT becomes known to the bats. Mortality rates derived in long-term studies may therefore have been underestimated. More than half of all European bat species inhabit maternal roosts in buildings, while other species hunt preferably at street lamps, in gardens, or cattle sheds [19]. Given this inherent overlap between SWT sites and sites frequently used by bats, studies addressing the initial collision risk of bats with SWTs are urgently needed.

The aim of our study was to simulate a worst-case scenario to estimate the maximum initial damage that an SWT could impose on the local bat fauna. Specifically, we were interested in the number of exploratory flights, rotor passes, and subsequent collisions of bats when confronted with SWTs newly installed in bat hotspots. We therefore sequentially placed a mobile SWT at six different sites of high bat activity, i.e., located close to vegetation or other structures of interest, and then monitored bat behavior for five consecutive nights using a stereo-optical infrared camera system and acoustic records. By conducting an operational experiment with the SWT at different rotor speeds we also provide knowledge on how bat behavior around SWTs is influenced by the different operational modes of the turbine. In addition, we examined species-specific differences in maintaining distance from the SWT, in order to gain further information on the respective differences in the collision risk.

## Material and methods

### Study sites

Six sites with high bat activity were selected based on their proximity to roosts, commuting structures along which bats commute between roosts and foraging sites, or highly frequented foraging sites. All of the sites were located in southwestern Germany, between the city of Freiburg and the Rhine River (Fig 1), and separated from each other by a distance of up to 40 km.

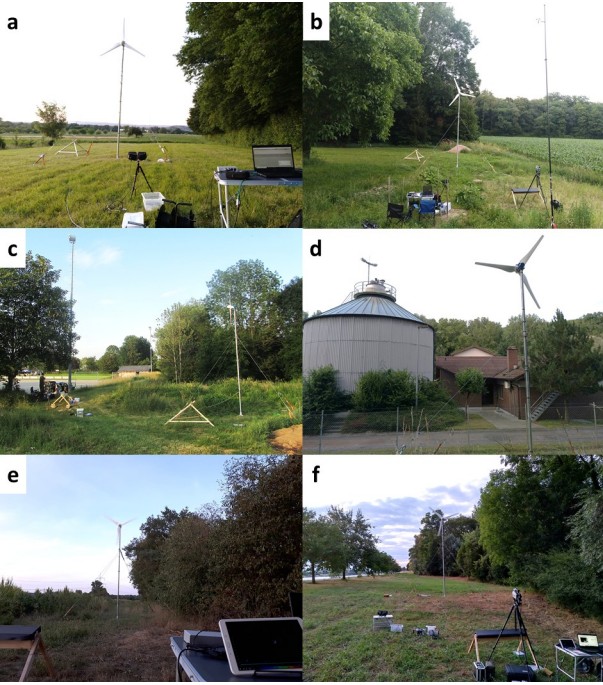

**Fig 1. Site selection.** Overview of the six sites (a–f) used in the experimental set-up.

The landscape structures at the sites included forest edges, hedgerows, riparian forests, and meadows of high insect density.

## Experimental design

Between May and August 2018, an operational experiment was conducted at each of the six sites during five consecutive evenings/nights, from 0.5 h before to 2.5 h after sunset. On the first night, bat activity was monitored acoustically and visually in the absence of a SWT. Instead, a reference telescope pole set up at the same position allowed acoustic monitoring at the same height (4 m) as subsequently used for the SWT but also served as a reference in distance measurements of bat flight paths during the first night. On all following nights, a mobile SWT (turbine type WKA 600, EAN 4251108115622, HeuSa GmbH, rotor diameter 2.8 m, nacelle height 6.1 m) was set up, separated from the relevant vegetation structures or buildings by only a few meters (Fig 1). The generator was replaced by a motor (MS 803–0,75 kW-6pol-B3, JS Technik GmbH) to allow the manipulation of rotor speeds. On the second and third nights, the rotor was either propelled at full power or held in position by brakes. This was done to assess whether the collision risk was higher when the SWT was immediately operated at full speed than when the bats were allowed an initially safe encounter and were thus able to adapt to the newly introduced, unknown object. On the fourth night, the SWT was propelled at half of its full power and on the fifth night operational modes that had been prevented on the previous days due to adverse weather conditions were tested (for details on the experimental schedule, see S1 Table).

The speed of the rotor blades was 88 rpm and 102 rpm at half- and full-motor power without wind, respectively. Two further rotor speed classes resulted from additional acceleration by wind, such that the experiment consisted of six possible operational modes: 1) no SWT, 2) SWT out of operation, and 4–6) four different rotor speed classes (class 1: 60–90 rpm, 32–50

km/h at the outer blade tip, class 2: 96–115 rpm, 51–60 km/h, class 3: 116–135 rpm, 61–71 km/h, class 4: > 135 rpm, >72 km/h.)

In addition to video recordings, the rotor was constantly monitored by at least one person using a thermal camera. In case of a collision, the SWT was immediately halted and the experiment terminated ahead of schedule at the respective site. Thus, there was a risk of a maximum six bat casualties in total across all sites. This risk was acknowledged in the permit granted by the Regional Council Freiburg (05.04.2018, file reference: 55–8852.44/100), which also confirmed that our experiment complied with the federal law on nature protection of the European Union and that there would be no negative effects on the bats at the population level.

## Bat acoustic activity

Bat acoustic activity was monitored for use in species identification and was automatically recorded using two electret ultrasound microphones (Knowles FG, Avisoft Bioacoustics) placed at 4 m height on either the SWT or the telescope pole and facing opposite directions to allow the omnidirectional recording of bat calls. The latter were evaluated by combining automatic classification and manual post-validation [20] using the software BATscreen V 1.0.5 (bat bioacoustictechnology GmbH, Winkelhaid, Germany).

## Three-dimensional recording of bat activity using a stereo-optical infrared camera

Bat activity was recorded three-dimensionally using a stereo-optical system that is part of a multisensor array still under development [21–23]. The development goals of the system are the visual detection and localization of bats during extended periods of time while keeping the associated costs to a minimum. The two infrared-sensitive cameras employed in the study each contained a 5MP 1/4" CMOS sensor OmniVision OV5647 and were controlled by a single-board computer (Raspberry Pi). Images were obtained at a frequency of 15 per second and stored on usb flash drives. To ensure matching timestamps, the system was equipped with a GPS module (Adafruit Ultimate GPS HAT). The cameras were positioned at a distance of 11–14 m from the turbine footing. Two infrared spotlights (IR06/60 850NM, Indexa GmbH) were placed at a distance of 2–3 m away from but facing the SWT. The system allowed for the detection of bats up to 20 m away.

The three dimensional coordinates of the bat positions were calculated using a homogeneous DLT (direct linear transformation) method that included several triangulation and calibration steps (see [25] for details). Each flight trajectory (the unit of replication used in the statistical analysis) consisted of all coordinates of the bat between its first and last appearance in the filmable cone-shaped area around the SWT. Different flight trajectories therefore do not necessarily correspond to different individuals. The minimum distance to the rotor was calculated as the point of each trajectory with the shortest distance to the area swept by the rotor. For the reference pole, a hypothetical disk with the same diameter as the SWT rotors was used.

## Habitat and weather data

Wind speeds were measured every minute using wind loggers (PCE-ADL 11, PCE instruments) installed at nacelle height. Temperature and humidity were recorded every 15 min at a height of 1 m from the ground (EasyLog EL-USB-2, LASCAR Electronics). Brightness was derived from the global solar radiation at DWD (Deutsche Wetterdienst) station 1443, Freiburg. This parameter was used in the statistical models to investigate whether bats behaved differently before total darkness set in, when, in addition to echolocation, optical orientation was possible.

## Statistical analyses

Statistical analyses were conducted using R 3.5 (The R Foundation for Statistical Computing). Linear mixed effect models (function *lmer* from R-package *lme4* [24]) were constructed to evaluate the effect of operational state and global radiation on the minimum distance of each flight trajectory to the rotor blades. Global radiation was included to assess whether bats maintained a larger distance from the SWT close to twilight, when visual recognition was still possible, than after dark, when echolocation was the sole means of orientation. SWT site was included as a random factor to account for seasonal and site-specific differences in bat activity and species composition. The models were evaluated visually by plotting their residuals against the fitted values and the residual quantiles against the quantiles of a normal distribution. Precipitation, wind speed, and temperature were removed from the final model because of their low variation in the data (mostly no wind and temperatures between 15 and 20˚C) and the lack of data at one site, which would have prevented a joint analysis (prior models including environmental parameters showed similar results). The effect of species or species group (e.g., *Myotis*) on the minimum distance was analyzed for a subset of flight trajectories for which correct species identification was possible.

## Results

### Bat species and activity levels

Species composition differed markedly between sites depending on the roosts or hunting sites in their vicinity. Commuting activity of 20 and 50 *Pipistrellus pipistrellus* individuals was observed at two sites (a and c), and of >20 *Myotis mystacinus* individuals at one site (c). Site d was less than 20 m apart from a maternal roost of >300 individuals of *P. pygmaeus*, 20 individuals of *P. pipistrellus*, and <5 individuals of *P. nathusii*. The activity level of all three species was high, particularly that of *P. pygmeaus*. Sites b and e were characterized by the high hunting activity of *M. myotis* and *M. natterer*i as well as *P. pipistrellus*. *Eptesicus serotinus* hunted frequently at site a, and *Nyctalus noctula* and *N. leisleri* sporadically at sites b, e, and f.

At the six sites, a total of 7,850 flight trajectories were filmed in high three-dimensional resolution during 84 h of video documentation (Fig 2). Since each bat could leave more than one flight trajectory by exiting and then re-entering the camera's image frame, the number of flight trajectories does not necessarily correspond to the number of filmed bat individuals. Among the recorded flight trajectories, 90% were within 10 m distance to the nacelle.

### Bat behavior around the rotor blades

High bat activity levels (7,850 flight trajectories) were documented within close distance to the SWT (10 m radius). Exploratory behavior was frequently observed, with many bats appearing to deviate from their original flight trajectory (even while commuting) to approach the rotor. The minimum distance of the filmed trajectories to the rotor varied between 0 and 18 m, with a mean of 4.6 m (± 2.6 m) across all sites. The results of the linear mixed model showed that, compared to the distance to the reference pole (7.78 m; 95%CI: 6.6–9 m), the bats approached the rotor 0.4 m closer if it was out of operation (95% confidence interval [95%CI]: 0.3–0.6 m) and 0.3 m closer if it was moving slowly (rotor velocity class 1, 95%CI: 0.1–0.4 m) (Fig 3 and Table 1). The distance of the bats to the rotor when operating at higher velocities was similar to that maintained to the reference pole (0.06–0.11 m). The bats remained farther away from the SWT as long as they could still see it than when they had to use echolocation to navigate in increasing darkness.

The number of paths that crossed the rotor diameter was calculated for each operational mode (Table 2). For the telescope pole, a hypothetical rotor was substituted. When the SWT

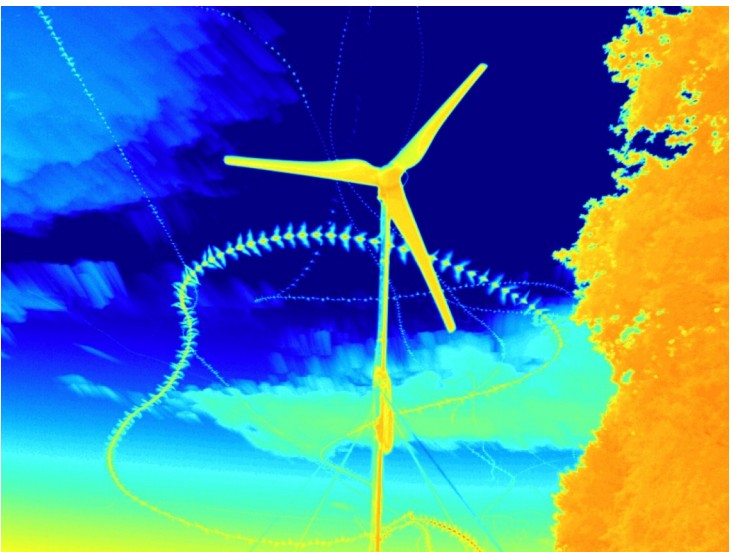

**Fig 2. Example of a bat passing the rotor blades of a small wind turbine at close distance.**

was out of operation and the rotor blades were not moving, between 2 and 46 flight trajectories crossed the rotor, corresponding to 1–8% of all documented trajectories. When the rotor was in motion, the number of crossing trajectories decreased to 0–6, depending on the rotor velocity and the site. At four of the six sites, no rotor passes were documented with the rotor operating at low speed (rotor class 1), but at five of six sites sporadic crosses occurred even at a high rotor speed. Overall, of the 7,850 documented trajectories, 176 crossed the rotor, including 65 while the rotor was in motion.

The sites differed markedly in vegetation structure, vicinity to roosts, commuting structures, and species range. This variation was mirrored in the results of the statistical analyses, in

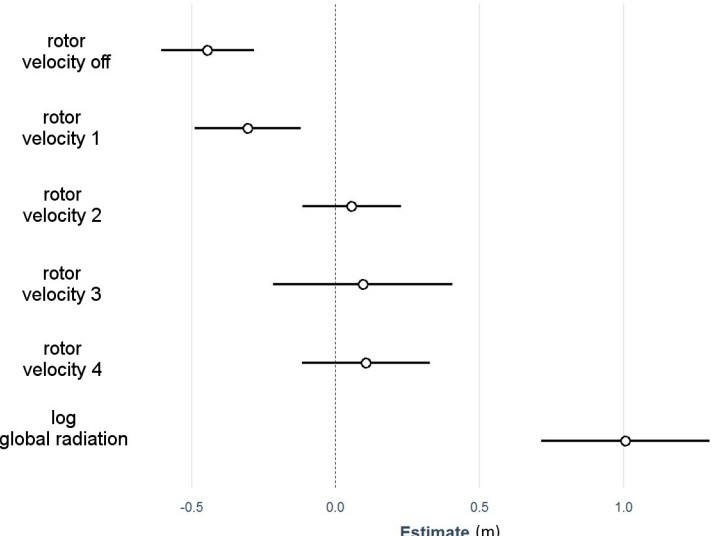

**Fig 3. Results of the linear mixed model assessing the minimum distance to the rotor dependent on the operational state of the SWT and the global radiation.** The monitoring site served as a random factor. Circles indicate regression coefficients, and lines the 95% confidence intervals. The telescope pole (dotted line) was set up as a reference.

**Table 1. Test statistics for Fig 3.**

|  | Estimate | Lower 95% CI | Upper 95% CI |
|---|---|---|---|
| Ø telescope pole | 7.78 | 6.56 | 8.99 |
| SWT no rotor movement | -0.45 | -0.61 | -0,28 |
| Rotor velocity 1 | -0.31 | -0.49 | -0.12 |
| Rotor velocity 2 | 0.06 | -0.12 | 0.23 |
| Rotor velocity 3 | 0.09 | -0.22 | 0.41 |
| Rotor velocity 4 | 0.11 | -0.12 | 0.33 |
| Log global radiation | 1.01 | 0.71 | 1.30 |

N = 7.850, $R^2$ fixed effects = 0.01, $R^2$ random effects = 0.17.

which site, as a random factor, accounted for more variation ($R^2$ = 0.17) than did the fixed effects ($R^2$ = 0.01). The influence of species composition became apparent in a separate analysis of a subset of flight trajectories (n = 1,158) by species that could be identified acoustically:

Hunting or commuting by individuals of the *Myotis* group is generally more structure-oriented than hunting by pipistrelles or even the Nyctaloid group, which preferably hunts in open space. This was reflected in our species-specific model, which showed that *Myotis* individuals approached the rotor 0.5 m closer than *Pipistrellus* individuals whereas Nyctaloid individuals remained 0.6m farther away than pipistrelles (Fig 4 and Table 3).

## Collision risk

One collision, of a *P. pygmaeus* individual, was documented at site e on November 8, 2018 at 23:30, during the fifth night of the experiment and shortly before the official schedule ending (S1 Video). The rotor blades were moving at ~45 km/h (speed of rotor blade tip, rotor speed class 1). The bat, which presumably died on impact, was catapulted into the upper ends of the nearby hedgerow (see Supplementary material). Unfortunately, it could not be located, despite immediate extensive search efforts using flashlights and thermal cameras. Searches early the next morning were also fruitless, possibly because the cadaver had already been removed by a predator.

## Discussion

### Bat activity levels and behavior

Bat activity levels were high at all six study sites in Southwestern Germany, with a total of 7,850 bat flight paths in close proximity to the rotor filmed during 5 days. By contrast, an operational experiment in Northern Germany conducted at 20 SWTs of ~20 m height reported

**Table 2. Flight trajectories crossing the rotor blade diameter for the different operational modes and sites.**

| Site | No SWT | SWT out of operation | Rotor velocity class 1 | Rotor velocity class 2 | Rotor velocity class 3 | Rotor velocity class 4 | Sum |
|---|---|---|---|---|---|---|---|
| a | 9/4.1 | 2/1.6 | 5/3.8 | 9/6.4 | 1/7.7 | 1/1.6 | **27/4.2** |
| b | 2/4.1 | 46/8.2 | 0 | 0 | 6/4.3 | 2/4.1 | **56/3.5** |
| c | 9/2.6 | 11/1.9 | 13/1.5 | 14/1.7 | 1/1.5 | 2/0.4 | **50/1.6** |
| d | 3/2.8 | 2/1 | 0/0 | 3/1.6 | 4/5.7 | 0/0 | **12/1.9** |
| e | 7/1.3 | 8/1.9 | 0 | 0 | 0 | 0 | **15/0.5** |
| f | 5/2.4 | 7/2.1 | 0/0 | 4/1.6 | - | - | **16/1.5** |
| Sum | 35/2.9 | 76/2.8 | 18/0.9 | 30/1.9 | 12/3.8 | 5/1.2 | **176/2.2** |

The absolute number/percentage of all recorded flight paths is shown.

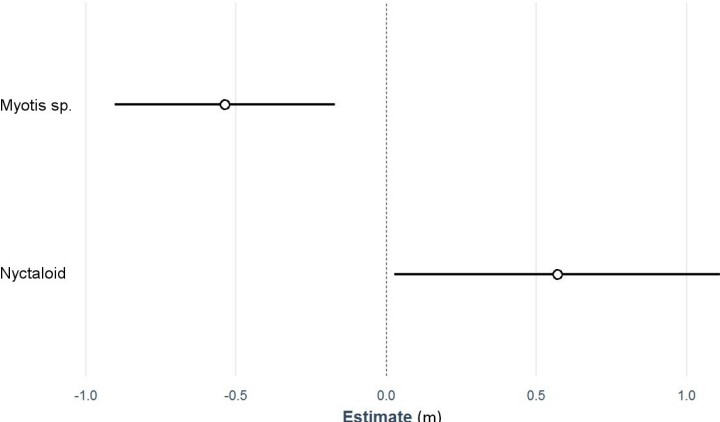

**Fig 4. Results of the linear mixed model assessing minimum distance to the rotor dependent on the bat species.**
The respective site served as a random factor (n = 1,158). Circles indicate regression coefficients, and lines the 95% confidence intervals. The *Pipistrellus* group (dotted line) was defined as the reference.

~500 flight paths during 24 nights with only 41 of the flight path approaching the rotor closer than 10 m [25]. Bats showed little exploratory behavior in the Northern Germany experiment and only one rotor pass (when the SWT was out of operation) whereas our recordings revealed frequent exploratory behavior and 176 rotor passes (65 while the rotor blades were in motion).

The striking differences between the two studies can be attributed to several factors: First, both bat activity levels and species number seem to be lower at SWT sites in Northern Germany than at those in other regions in Germany, where a larger number of acoustic recordings have been documented using the same technical settings [25,26]. Whereas Nyctaloids, especially *E. serotinus*, dominated in the Northern Germany study, the sites in Southwestern Germany chosen for placement of the mobile SWT were close to the roosts and commuting structures of more structure-oriented species such as those of *Myotis* or *Pipistrellus*. Our results showed that the general structural affinities of these groups are reflected by the distance from the rotor maintained by their individual members. Second, the SWT in our study was only half the height of the SWT assessed in Northern Germany (~ 20 m). The larger overlap of SWTs with the activity range of most bats, especially those within the structure bound *Myotis* and *Pipistrellus* groups, would result in much higher contact probabilities. Third, and most important, the sites in Southwestern Germany were explicitly chosen for their high bat activity levels and thus simulated a worst-case scenario. The much higher bat activity levels in our study than in the study from Northern Germany were therefore not surprising, although the 200-fold difference was unexpected.

The higher rate of exploratory behavior can also be explained by our having documented bat behavior directly after installation of the SWT, whereas in Northern Germany the SWT had already been in operation for several years. Bats tend to be curious about new objects,

**Table 3. Test statistics for Fig 4.**

|  | Estimate | Lower 95% CI | Upper 95% CI |
|---|---|---|---|
| Ø minimum Distance Pipistrelloids | 4.57 | 3.75 | 5.39 |
| Myotis sp. | -0.54 | -0.90 | -0,17 |
| Nyctaloids | 0.57 | 0.03 | 1.11 |

N = 1,158, $R^2$ fixed effects = 0.01, $R^2$ random effects = 0.17.

which would presumably include a SWT newly positioned in a formerly undisturbed area. However, in contrast to the SWT, the telescope pole was not explored, probably because its long, thin structure resembled that of vegetation such as dead trees or other common landscape features. Bats approached the rotor 0.4 m (95%CI: 0.3–0.6 m) closer than they did the reference pole when the blades were moving slowly, an indication that moving objects are of greater interest than immobile objects. At a higher rotor velocity, the distance maintained by the bats to the SWT and the reference pole was similar. This behavior may have been due to a disruption of echolocation calls with increasing rotor speed [27,28], which hindered a closer examination by the bats.

## Collision risk

Among the 7,850 flight paths documented in close proximity to the SWT, only one resulted in a collision. The number of flight paths that crossed the rotor blades was small (n = 176), and only a third of those were documented when the rotor was in motion. However, over the 5 days of the experiment, the rotor blades were moving for a total of only 47 h. Thus, although all but one of the bats that crossed the moving rotor blades during the study were able to avoid a collision, longer experimental time frames and a continuously running SWT would likely have resulted in a larger number of fatalities. A relatively low risk of collision between bats and SWTs was reported in a 2-year study, during which time no carcasses were found at 21 SWTs [16], whereas in a 3-year study up to three carcasses were found at 31 SWTs [15]. A questionnaire study involving 271 SWT owners yielded reports of three carcasses in total, corresponding to an estimated annual rate of 0.008–0.169 collisions per SWT [16]. Carcass searches with trained dogs delivered a similarly low mean collision rate of 0.81 dead bats per SWT/year, but in a worst-case scenario the rate may be as high as 15 dead bats/year per SWT [15]. A Swiss study from 2016 reported eight collisions between July and September at three SWTs located in an industrial area [29], thus demonstrating that these worst-case scenarios can become reality. As in the Swiss study, the collision in our experiment occurred at the end of the maternal roost phase, when newborn bats begin to accompany adults on their flights. The killed bat may therefore have been a newborn that lacked the maneuverability of an adult, but since the carcass could not be located this could not be confirmed.

It should also be pointed out that to conduct a worst-case experiment we explicitly chose high-risk sites, i.e., those with exceptionally high bat activities. In addition, SWTs would normally be placed at greater distance to buildings or vegetation, for better wind exposure. Moreover, the rotor blades of our SWT were accelerated during calm nights by a motor, which would not be done under real-life operating conditions. Accordingly, the collision risk at standard SWT sites will most likely be lower than in our study.

## Methodological considerations and study limitations

The transferability of our results to other local and structural conditions, bat species, and SWT types is limited. Despite the generally broad species richness in our study, with at least nine different species from the pipistrelle, Nyctaloid, and *Myotis* groups, rare species such as the barbastelle bat were not assessed. Also, due to logistical limitations, our mobile SWT, with its height of only 10 m and a rotor diameter of 3 m, was one of the smallest SWTs available. An extrapolation of our findings to larger SWTs or other SWT types such as those with vertically moving blades is not possible.

Interestingly, much of the variation in our data was explained by the SWT site, which differed with respect to vegetation type, vegetation height, bat activity pattern (roost, commuting, hunting), bat species, and landscape variables (proximity to settlements, forests, or agricultural

land). Further controlled experiments focusing on the impact of single-site factors would improve the predictability of bat activity and collision risks at SWTs.

## Conclusion

Our study showed that SWTs situated near roosts, commuting or hunting structures will result in high bat activities and exploratory behavior within close proximity to the rotor. This will not necessarily lead to high casualty rates, as most bats in our study were adept at avoiding the moving rotor blades. However, the collision of one *P. pygmeaus* individual at one of the six experimental sites demonstrates that caution is warranted, especially if the potential cumulative effects over long operational periods are considered.

Although environmental impact assessments are typically not required for SWTs, their operation must still comply with the federal law on nature protection of the European Union; that is, SWT operation must not pose an increased mortality risk for bats. While our study design provoked a higher collision risk than would be expected under standard SWT conditions, the threshold currently applied to large wind turbines of 1–2 dead bats per wind turbine and year could conceivably also be reached by SWTs. However, as SWTs have a much lower energy output than large wind turbines and their cost-benefit is accordingly lower, specific regulations that take these factors into account are needed.

We therefore suggest bat-friendly SWT site selection, in which proximity to structures of particular interest to bats, such as buildings, trees, forest edges, hedgerows, and waterbodies are avoided. If this is not possible, such as with SWTs designed for installation on rooftops, curtailment algorithms during periods of high bat activity will decrease the collision risk for bats at SWT sites.

## Supporting information

**S1 Table. Overview over the different operational modes of the turbine applied to each site.** * = Experiment stopped ahead of schedule due to bat collision.
(DOCX)

**S1 Video. Documentation of the collision of *P. pygmaeus* during the experiment.**
(AVI)

## Acknowledgments

We thank Dr. Hendrik Reers for his input in the experimental design and set-up of the mobile SWT, Fränzi Korner-Nievergelt from Oikostat.ch for statistical advice, and all site owners for their cooperation in this study.

## Author Contributions

**Conceptualization:** Stefanie A. Hartmann, Horst Schauer-Weisshahn, Robert Brinkmann.

**Data curation:** Klaus Hochradel, Sören Greule, Felix Günther, Bruntje Luedtke, Horst Schauer-Weisshahn.

**Formal analysis:** Klaus Hochradel, Sören Greule, Felix Günther.

**Funding acquisition:** Stefanie A. Hartmann, Robert Brinkmann.

**Investigation:** Stefanie A. Hartmann, Sören Greule, Bruntje Luedtke, Horst Schauer-Weisshahn, Robert Brinkmann.

**Methodology:** Stefanie A. Hartmann, Klaus Hochradel, Sören Greule, Bruntje Luedtke, Horst Schauer-Weisshahn.

**Project administration:** Stefanie A. Hartmann.

**Resources:** Klaus Hochradel, Horst Schauer-Weisshahn.

**Software:** Klaus Hochradel, Sören Greule, Felix Günther.

**Supervision:** Robert Brinkmann.

**Validation:** Klaus Hochradel, Sören Greule, Felix Günther, Robert Brinkmann.

**Visualization:** Klaus Hochradel, Sören Greule, Felix Günther.

**Writing – original draft:** Stefanie A. Hartmann.

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
