## [Decision Letter · Decision Letter 0]

13 Apr 2021

PONE-D-20-40193

Collision risk of bats at small wind turbines- worst-case scenarios near roosts, transfer or hunting structures

PLOS ONE

Dear Dr. Hartmann,

Thank you for submitting your manuscript to PLOS ONE. After careful consideration, we feel that it has merit but does not fully meet PLOS ONE’s publication criteria as it currently stands. Therefore, we invite you to submit a revised version of the manuscript that addresses the points raised during the review process.

We look forward to receiving your revised manuscript.

Kind regards,

Brock Fenton

Academic Editor

PLOS ONE

Journal Requirements:

2. Please ensure that you refer to Figures 1-4 in your text as, if accepted, production will need this reference to link the reader to each figure.

3. We note you have included a table to which you do not refer in the text of your manuscript. Please ensure that you refer to Table 1 in your text; if accepted, production will need this reference to link the reader to the Table.

Additional Editor Comments:

Dear Stefanie Hartmann:

Thank you for submitting your manuscript. You will see that the reviewers are at odds about the manuscript. I would be grateful if you would carefully consider the points raised by the more critical of the reviewers and adjust your manuscript accordingly. If you choose to follow this route, I will send the revised version back to the more critical reviewer and to another reviewer.

I look forward to hearing form you

thanks

Brock Fenton

Reviewers' comments:

Reviewer's Responses to Questions

**Comments to the Author**

1. Is the manuscript technically sound, and do the data support the conclusions?

Reviewer #1: Yes

Reviewer #2: Partly

2. Has the statistical analysis been performed appropriately and rigorously? 

Reviewer #1: Yes

Reviewer #2: I Don't Know

3. Have the authors made all data underlying the findings in their manuscript fully available?

Reviewer #1: Yes

Reviewer #2: Yes

4. Is the manuscript presented in an intelligible fashion and written in standard English?

Reviewer #1: No

Reviewer #2: Yes

5. Review Comments to the Author

Reviewer #1: 1. English needs a thorough review. Grammatical errors and inelegancies are common and must be fixed.

2. Remove speculations and lengthy details

3. Why no comparison with the results of the British studies?

Reviewer #2: Summary

In this experiment, the authors examined the proximity to which bats approach small wind turbines (SWT) by temporarily installing a turbine at six different sites and employing different operational treatments. I think the key interesting result here is that bats appeared to approach SWT more closely than a pole used at the same site as a reference point, but only when it was not operational at all, or the blades were moving slowly. However, whilst the authors discuss the results of the treatment where the blades were moving slowly, I couldn’t see any arguments for why a turbine and a vertical pole were approached differently if the blades were not moving at all – it would be interesting to consider what kind of behavioural mechanism may result in this finding.

Whilst I felt there were some interesting results in this study, I have a number of concerns, primarily:

• The relevance of the study to realistic installation scenarios. Were the installations following any guidelines? From the photos provided some of these locations did not look very suitable for turbines (in terms of laminar flow and maximal wind speed), and are not in accordance with the Eurobats guidance on locating small wind turbines.

• The wording of the aims are very vague (L82 – 89) – it needs to be clearer what is actually being measured that will enable the aim to be addressed. What does damage mean? Also, I think the bats’ response is going to vary so much by scenario there's no way of knowing whether this is max damage or not.

• Results felt quite fragmented in places and lacking key information – findings from a linear model are presented but with no test statistics and with the results spread over a large area e.g. the headline result is given on one page but we don’t find out until the following page that the ability of the model to explain the variation in the dataset is tiny (R2 of 1%).

• The final conclusion (L28-30) of the abstract is that casualties can reach relevant levels over time. I do not feel the authors have the data to make this claim, and no information is presented on what constitutes a “relevant level”. Bats were monitored behaviour over 5 days in areas of purposefully high activity – we don’t know what would have happened if the turbines had been in place for longer.

Ethics

I found it very surprising that the institute the researchers are based at does not have a process for reviewing potentially harmful research activities. The fact that a bat was killed as a result of this work demonstrates this – this isn’t really a criticism of the nature of the research as there’s clearly a role for these sorts of experiments. However, there is certainly a conversation to be had about what is acceptable risk and what might be considered unethical. Readers may take a variety of views so some consideration in the discussion (or an expansion on the existing text), over and above “it was legal” is warranted.

General

Detailed comments

L24: The term reference pole is introduced here with no context – a little rephrasing here would clarify the purpose of the pole.

L31: The last line of the abstract felt a bit abrupt – a little rephrasing/expansion/clarification is needed.

L67: the authors consider that a major shortcoming of previous research is that the turbines had been up for longer time periods than in this study, meaning that the immediate effects could not be addressed. I agree that it’s really interesting to look at whether effects change over time but feel that this is only a shortcoming if your main interests are quantifying short term effects. If you’re interested in what happens over the longer term, this is not a short coming at all. I would rephrase to reflect this.

L74: is there a ref to back up the statement that the “preferred installation [of SWT] in close

proximity to the buildings which they supply with energy”? Suitable locations for SWT need to avoid areas where barriers (such as buildings, treelines) can disrupt wind flow. Indeed this is why turbines are no longer installed in urban areas because the pattern of wind flow is not suitable. Most of the SWT I am aware of today are installed in fields, away from buildings.

L82: not sure “unravel” is the most appropriate word to use here – quantify/characterise?

L85: change acoustical to acoustic

L92: I am not familiar with the term “transfer structures” (and at L177 “transfer activity”) – some clarification or rephrasing needed. I presume this means commuting to foraging sites?

L100: We need more information on the expt set up e.g. distance to buildings/vegetation etc. How far away was the reference pole to the turbine?

L128: We need some information on what these acoustic data were used for? I was expecting to see an analysis of activity at the turbine vs. ref pole, but I think it was to obtain species id for some of the trajectories?

L138: We need information here on the metrics being quantified e.g. minimum distance

L160: It would be helpful if the stats could refer back to the specific objectives being addressed and in the same order as the introduction. Is the unit of replication here the flight trajectory? Some discussion that there will inevitably be pseudo-replication in the analyses since it will not be possible to distinguish individuals. Also, in the results there is a comparison of the turbine and the ref pole but there is no explanation here on how or why they are doing this.

L164 & 203: these sentences sounds awkward and need rephrasing

L194: Was there a reason for picking 10m here as the distance? I realise any distance picked might be arbitrary but if there was a rationale for the 10m it would be useful to clarify here.

L197: These are the key results but there is much which I found unclear here. Are these - are these modelled estimates or raw data? Where are the results of the model and the test statistics? We find out on the next page that R2 for fixed effects is only 1% - I found this quite hard to reconcile with the confidence intervals shown in Fig 3.

L215: the number of bats passing in front of the rotor when it’s not moving doesn’t seem like a very relevant metric?

L225: Some of the sites that differed in vegetation structure etc also differed in species composition so I just don’t feel there is sufficient replication to tease this all out

L265: I’m not familiar with the term “structure bound” for bats? Pipistrellus sp are also known to forage along forest edge and interiors, and are closely associated with buildings.

L279 (and abstract): these point estimates should only be given with the confidence intervals.

L284: comment is made that no habituation over the 5 days was evident but I think it’s important to note that the treatments changed each night so this doesn’t seem very surprising.

L294: 64 of 65 trajectories (not bats)

L347: interestingly, this siting recommendation was also made by Minderman et al. but for opposite reasons (because of the apparent disturbance / avoidance effect detected in that study).

6. PLOS authors have the option to publish the peer review history of their article (what does this mean?). If published, this will include your full peer review and any attached files.

Reviewer #1: No

Reviewer #2: No

---

## [Author Response · Author response to Decision Letter 0]

7 May 2021

Dear Prof. Brock Fenton,

Thank you very much for the great opportunity to publish in Plos One! We are greatful for the constructive comments from you and both reviewers and address each point separately (see below, printed in black, for better readability we coloured your and the reviewers remarks in green- of course works only in the attached word document). We changed all formatting according to your comment and the figures have been checked with pace. We also sent our manuscript to a professional proof reading service to comply with Reviewer 1´s comment on the quality of our English. We have left also the English correction visible with track changes. Our proofreader suggested also a different wording in the title, we hope that is okay? Otherwise please feel free to not accept the changes.

The changes we conducted as a response to the Reviewer´s comments are addressed with line numbers which correspond to the “track changes” manuscript version so they can be easier found. 

We hope our revised manuscript now merits a publication in PlosOne.

Best regards,

Stefanie Hartmann

PONE-D-20-40193

Collision risk of bats at small wind turbines- worst-case scenarios near roosts, transfer or hunting structures

PLOS ONE

Dear Dr. Hartmann,

Thank you for submitting your manuscript to PLOS ONE. After careful consideration, we feel that it has merit but does not fully meet PLOS ONE’s publication criteria as it currently stands. Therefore, we invite you to submit a revised version of the manuscript that addresses the points raised during the review process.

We look forward to receiving your revised manuscript.

Kind regards,

Brock Fenton

Academic Editor

PLOS ONE

Journal Requirements:

 Point adopted

2. Please ensure that you refer to Figures 1-4 in your text as, if accepted, production will need this reference to link the reader to each figure.

 Point adopted

3. We note you have included a table to which you do not refer in the text of your manuscript. Please ensure that you refer to Table 1 in your text; if accepted, production will need this reference to link the reader to the Table.

 Point adopted

Point adopted

All above formatting changes have been conducted without tracking changes for better readability.

Additional Editor Comments:

Dear Stefanie Hartmann:

Thank you for submitting your manuscript. You will see that the reviewers are at odds about the manuscript. I would be grateful if you would carefully consider the points raised by the more critical of the reviewers and adjust your manuscript accordingly. If you choose to follow this route, I will send the revised version back to the more critical reviewer and to another reviewer.

I look forward to hearing form you

thanks

Brock Fenton

Reviewers' comments:

Reviewer's Responses to Questions

Comments to the Author

1. Is the manuscript technically sound, and do the data support the conclusions?

Reviewer #1: Yes

Reviewer #2: Partly

2. Has the statistical analysis been performed appropriately and rigorously? 

Reviewer #1: Yes

Reviewer #2: I Don't Know

3. Have the authors made all data underlying the findings in their manuscript fully available?

Reviewer #1: Yes

Reviewer #2: Yes

4. Is the manuscript presented in an intelligible fashion and written in standard English?

Reviewer #1: No

Reviewer #2: Yes

5. Review Comments to the Author

Reviewer #1: 1. English needs a thorough review. Grammatical errors and inelegancies are common and must be fixed.

Thank you for your hint. Our manuscript has now been worked through by a professional native speaking proofreader (Line 1-469).

2. Remove speculations and lengthy details

We changed a few sentences according to your recommendation, we especially hope the English proofreading has helped this point throughout the entire manuscript. If you still feel we need to shorten some aspects, please let us know which exactly and we will do our best. 

3. Why no comparison with the results of the British studies?

We list the collisions reported in Minderman et al 2012 and 2015 as well as Moyle et al 2016, both from their systematic searches as well as questionnaires (Line 393-400, line numbers refer to the manuscript with track changes). We compare their conclusions with our results. If we have missed further British studies of relevance to our manuscript, please let us know so we can include those also.

However we include no comparison with respect to avoidance behaviour, the main focus of the mentioned British studies, as this is out of the scope of our study. Our technical approach covers only a 20m-radius around the SWT and we therefore cannot draw any conclusions about a general avoidance of the area around the SWT, which takes place up to several 100m distance according to the British studies. 

Reviewer #2: Summary

In this experiment, the authors examined the proximity to which bats approach small wind turbines (SWT) by temporarily installing a turbine at six different sites and employing different operational treatments. I think the key interesting result here is that bats appeared to approach SWT more closely than a pole used at the same site as a reference point, but only when it was not operational at all, or the blades were moving slowly. However, whilst the authors discuss the results of the treatment where the blades were moving slowly, I couldn’t see any arguments for why a turbine and a vertical pole were approached differently if the blades were not moving at all – it would be interesting to consider what kind of behavioural mechanism may result in this finding.

We think that the reason for this finding could be that the telescope pole is not such an conspicuos new structure as an SWT with three rotor blades is. Like, it very much resemble a thin dead tree and does not take in as much room as the blades. We now include this possible explication in the manuscript (L 366-369).

Whilst I felt there were some interesting results in this study, I have a number of concerns, primarily:

• The relevance of the study to realistic installation scenarios. Were the installations following any guidelines? From the photos provided some of these locations did not look very suitable for turbines (in terms of laminar flow and maximal wind speed), and are not in accordance with the Eurobats guidance on locating small wind turbines.

This is correct. We explicitly chose to simulate a worst-case scenario and therefore placed the SWT closer to the vegetation than recommended in the guidelines or economically reasonable. While we mention that we aimed at a worst-case scenario, we had not emphasized this aspect so far. It is now included in the manuscript both in the methods (L. 104, L. 133-135) as well as in the discussion (L. 411-413).

• The wording of the aims are very vague (L82 – 89) – it needs to be clearer what is actually being measured that will enable the aim to be addressed. What does damage mean? Also, I think the bats’ response is going to vary so much by scenario there's no way of knowing whether this is max damage or not.

You are right. We now specifiy that we are interested in the amount of exploratory flights, rotor passes and subsequent collisions. We also changed our wording so that it becomes clearer we only aim to approach the maximum damage but of course we cannot claim to have reached this aim (L 101-111).

• Results felt quite fragmented in places and lacking key information – findings from a linear model are presented but with no test statistics and with the results spread over a large area e.g. the headline result is given on one page but we don’t find out until the following page that the ability of the model to explain the variation in the dataset is tiny (R2 of 1%).

We now present the findings more concisely and included a table with the test statistics (Table 1, L. 290-297)

• The final conclusion (L28-30) of the abstract is that casualties can reach relevant levels over time. I do not feel the authors have the data to make this claim, and no information is presented on what constitutes a “relevant level”. Bats were monitored behaviour over 5 days in areas of purposefully high activity – we don’t know what would have happened if the turbines had been in place for longer.

You are right. We now adjusted the wording for a more precise statement and to better match our conclusions (L. 34-41).

Ethics

I found it very surprising that the institute the researchers are based at does not have a process for reviewing potentially harmful research activities. The fact that a bat was killed as a result of this work demonstrates this – this isn’t really a criticism of the nature of the research as there’s clearly a role for these sorts of experiments. However, there is certainly a conversation to be had about what is acceptable risk and what might be considered unethical. Readers may take a variety of views so some consideration in the discussion (or an expansion on the existing text), over and above “it was legal” is warranted.

We had intense communications with the Regional Council, which in our case is responsible for reviewing potentially harmful research activities. We first had tried to prevent bats from entering the rotor with nets, however this did not lead to realistic results as bats were able to cross the rotor unharmed (and we would not have been able to differentiate from real collisions) and also bats approached the rotor from all sites and in 13 out of 14 flights avoided the net in last second (which left us unknowing whether they avoided the rotor or the net). We thus finally agreed on an experimental setup accepting a small number of killed bats (1 per site, at first collision the experiment was to be stopped immediately). We knew- in case of higher collision rates- that this approach would not let us properly calculate the number of collisions per day/week, but that was the compromise to not harm the local populations any further. We supplemented the respective information with the biological component (additonal to the legal part):

„In case of a collision, the SWT was immediately halted and the experiment stopped ahead of schedule at the respective site. Thus, we ran a risk of maximum six bat casualties in total across all sites, for which we had obtained a permit of the Regional Council Freiburg (05.04.2018, file reference: 55-8852.44/100) to ensure our experiment complies with the federal law on nature protection of the European Union and to avoid negative effects on population level.“

If you consider it helpful to include further details or a repetition of the above aspects in a different part of the manuscript, please let us know.

General

Detailed comments

L24: The term reference pole is introduced here with no context – a little rephrasing here would clarify the purpose of the pole.

You are right, we now formulate it more generally in the abstract and provide more detail about the reference pole in the manuscript itself (L128-132).

L31: The last line of the abstract felt a bit abrupt – a little rephrasing/expansion/clarification is needed.

Thanks for your comment, we now reworded the last two sentences, see above.

L67: the authors consider that a major shortcoming of previous research is that the turbines had been up for longer time periods than in this study, meaning that the immediate effects could not be addressed. I agree that it’s really interesting to look at whether effects change over time but feel that this is only a shortcoming if your main interests are quantifying short term effects. If you’re interested in what happens over the longer term, this is not a short coming at all. I would rephrase to reflect this.

Thank you for the suggestion, point adopted (L81-84). 

L74: is there a ref to back up the statement that the “preferred installation [of SWT] in close

proximity to the buildings which they supply with energy”? Suitable locations for SWT need to avoid areas where barriers (such as buildings, treelines) can disrupt wind flow. Indeed this is why turbines are no longer installed in urban areas because the pattern of wind flow is not suitable. Most of the SWT I am aware of today are installed in fields, away from buildings.

In Germany we find most SWT closer than 150 m to the respective buildings. All 10 SWTs in our first study (reference 25) had less than 100 m distance. However as this appears to be a regional or national finding, we therefore decided to rephrase our sentence to a more general comparison with large wind turbines (L91-96).

L82: not sure “unravel” is the most appropriate word to use here – quantify/characterise?

We are sorry for our apparently improvable english! We now have sent the manuscript to proofreading prior to resubmission. In this specific case, the wording has been changed already in response to a prior comment.

L85: change acoustical to acoustic

Thank you, point adopted.

L92: I am not familiar with the term “transfer structures” (and at L177 “transfer activity”) – some clarification or rephrasing needed. I presume this means commuting to foraging sites?

Yes, the term is now specified (L 115-117). If „commuting structure“ or „commuting activity“ is the official term, we can also substitute „transfer“ throughout the entire manuscript?

L100: We need more information on the expt set up e.g. distance to buildings/vegetation etc. How far away was the reference pole to the turbine?

The reference pole was present only during the first night, right at the same position where the SWT was to be standing during the following nights. It served as a reference for distance measurements and to allow acoustic recording in the same height (4 m) as with SWT. We see this point has not been adequately described in the first version of the manuscript and now improved it, as well as providing additional information on the distance to vegetation etc (L128-136).

L128: We need some information on what these acoustic data were used for? I was expecting to see an analysis of activity at the turbine vs. ref pole, but I think it was to obtain species id for some of the trajectories?

Yes, point adopted (L. 165).

L138: We need information here on the metrics being quantified e.g. minimum distance

Point adopted (L. 188-196).

L160: It would be helpful if the stats could refer back to the specific objectives being addressed and in the same order as the introduction. Is the unit of replication here the flight trajectory? Some discussion that there will inevitably be pseudo-replication in the analyses since it will not be possible to distinguish individuals. Also, in the results there is a comparison of the turbine and the ref pole but there is no explanation here on how or why they are doing this.

With the rephrased introduction, the match between the aims and the presented statistics should now be improved.

Yes, flight trajectory is the unit of replication, we now provide further information on this aspect and also how distance was calculated (L. 188-196). 

Concerning pseudoreplication: Yes, you are right, some individuals will have entered the filmable area more than once. However given the very high number of individuals due to the chosen hot-spots, most flightpaths will belong to different individuals. It would also simply not have been feasible to individually assign each flightpaths to individuals because individuals were not marked or identifiable. Also we think individual differences in approaching the SWT are neglible compared to the differences in species or due to hot-spot-type (transfer, hunting or roost). However we now acknowledge this point in the manuscript.

The aspects concerning the reference pole should become clear now with the additional information provided to answer a prior comment. 

L164 & 203: these sentences sounds awkward and need rephrasing

Point adopted (L. 213-216).

L194: Was there a reason for picking 10m here as the distance? I realise any distance picked might be arbitrary but if there was a rationale for the 10m it would be useful to clarify here.

It was chosen for a better comparison with a previously conducted study (where it was chosen arbitrarily).

L197: These are the key results but there is much which I found unclear here. Are these - are these modelled estimates or raw data? Where are the results of the model and the test statistics? We find out on the next page that R2 for fixed effects is only 1% - I found this quite hard to reconcile with the confidence intervals shown in Fig 3.

We have closely collaborated with a professional statistical company (oikostat.ch, see also Acknowledgements) who confirmed all our statistical tests and interpretations for correctness.

The confidence interval is mainly determined by the sample size, whereas the R2 is a measure of the effect size in relation to the overall variance in the data. Therefore narrow C.I.s do not contradict the fact that our fixed effects explain only little variation. We have a large sample size (n=7850) compared to the variance. The effect sizes are clear (small uncertainty), but small compared to the variance in the raw data. This means that beside the effects we describe, there are also other factors determining the variance in the data. 

L215: the number of bats passing in front of the rotor when it’s not moving doesn’t seem like a very relevant metric?

We think it still indicates the amount of curiosity bats show towards the blades in general. It is no indicator for collision risk, however. 

L225: Some of the sites that differed in vegetation structure etc also differed in species composition so I just don’t feel there is sufficient replication to tease this all out

You are right, it is impossible to derive which of the different factors have a major or minor contribution to the site differences. We changed the phrases accordingly (L 298-299).

L265: I’m not familiar with the term “structure bound” for bats? Pipistrellus sp are also known to forage along forest edge and interiors, and are closely associated with buildings.

Point adopted, our proof reader suggested to switch to “structure-oriented” which we did (L.300, L.345).

L279 (and abstract): these point estimates should only be given with the confidence intervals.

Point adopted.

L284: comment is made that no habituation over the 5 days was evident but I think it’s important to note that the treatments changed each night so this doesn’t seem very surprising.

You are right. We could have expected to see a difference only in case it had come to collisions already on day 2 of those SWT which started with full motor speed (compared to those who had moving motor blades on day 3, after a day of habituation). This was our original idea for why we used this experimantal design. But as we had only one, and late (day 5) collision, drawbacks on habituation make little sense in this case. We there fore removed the respective phrase (L. 376).

L294: 64 of 65 trajectories (not bats)

I am sorry, of course you are right. Thanks! As we want to refer to individuals (not trajectories) in our argument, we rephrased accordingly by replacing the specific numbers with „most bats“(L.388).

L347: interestingly, this siting recommendation was also made by Minderman et al. but for opposite reasons (because of the apparent disturbance / avoidance effect detected in that study).

Both mortality and disturbance are negative influences of SWT on local populations, which can be minimized with increasing distances between SWT and bat hot-spots. We hope consistent recommendations across a variety of studies help to implement respective avoidance measures. Thank you for your constructive comments on our manuscript!!

6. PLOS authors have the option to publish the peer review history of their article (what does this mean?). If published, this will include your full peer review and any attached files.

Do you want your identity to be public for this peer review? For information about this choice, including consent withdrawal, please see our Privacy Policy.

Reviewer #1: No

Reviewer #2: No

---

## [Decision Letter · Decision Letter 1]

17 May 2021

PONE-D-20-40193R1

Collision risk of bats with small wind turbines: worst-case scenarios near roosts, transfer paths, and hunting structures

PLOS ONE

Dear Dr. Stephanie Hartmann :

Thank you for submitting your manuscript to PLOS ONE. After careful consideration, we feel that it has merit but does not fully meet PLOS ONE’s publication criteria as it currently stands. Therefore, we invite you to submit a revised version of the manuscript that addresses the points raised during the review process.

ACADEMIC EDITOR:

Thank you for attending to the issues raised by the reviewer.  Note there there are a few other comments for you to consider.

thanks

Brock

We look forward to receiving your revised manuscript.

Kind regards,

Brock Fenton

Academic Editor

PLOS ONE

Journal Requirements:

Reviewers' comments:

Reviewer's Responses to Questions

**Comments to the Author**

1. If the authors have adequately addressed your comments raised in a previous round of review and you feel that this manuscript is now acceptable for publication, you may indicate that here to bypass the “Comments to the Author” section, enter your conflict of interest statement in the “Confidential to Editor” section, and submit your "Accept" recommendation.

Reviewer #2: (No Response)

2. Is the manuscript technically sound, and do the data support the conclusions?

Reviewer #2: Yes

3. Has the statistical analysis been performed appropriately and rigorously? 

Reviewer #2: Yes

4. Have the authors made all data underlying the findings in their manuscript fully available?

Reviewer #2: Yes

5. Is the manuscript presented in an intelligible fashion and written in standard English?

Reviewer #2: Yes

6. Review Comments to the Author

Reviewer #2: Thanks for your revisions, which I feel have greatly improved this ms. I do think the term "transfer activity/route" either needs defining earlier on or substituting with "commuting" which I think is more commonly used.

I did have a query about the comparison on the % of flights passing within 10 m of the turbine (ref 25). Using % as a metric is only a fair comparison if both recording set-ups have exactly the same field of view. In the methods you state that bats could be detected upto 20 m from the turbine. Was this the same with the study outlined in ref 25? Clearly, the narrower the field of view the the higher the proportion of flights will be recorded within 10m of the turbine.

7. PLOS authors have the option to publish the peer review history of their article (what does this mean?). If published, this will include your full peer review and any attached files.

Reviewer #2: No

---

## [Author Response · Author response to Decision Letter 1]

8 Jun 2021

Dear Brock Fenton, Dear Reviewer,

thank you very much for your answer!!

Let me first update the Funding and Competing Interest sections, below you will find the answers to the last revision.

Funding:

This project was funded by the German Bundesamt für Naturschutz, FGII 4.3/Naturschutz und erneuerbare Energien, Projekt zum Forschungs- und Entwicklungsvorhaben aus Mitteln des Bundesministeriums für Umwelt, Naturschutz und nukleare Sicherheit (BMU), Einzelplan 16, Kapitel 1604, Titel 54401. Förderkennzeichen: 3517860600

The funder provided support in the form of salaries for authors [S.A.H., S.G, F.G., B.L., H-S.-W, R.B.], but did not have any additional role in the study design, data collection and preparation of the manuscript. The decision to publish internationally was made together with the funder. The additional salary necessary for the preparation of the english manuscript as well as the revisions was covered by FrInaT GmbH.

Competing Interests statement

During the time of the conduction of the study, all authors except Klaus Hochradel have been affiliated with FrInaT GmbH. The project itself was based with FrInaT. Once the project was finished, two authors (S.G., F.G.) were hired by Oekofor GbR. Oekofor and FrInaT collaborate with several projects and have no competing interests in a financial, professional or personal aspect.

The commercial affiliations to FrInaT GmbH and Oekofor GbR do not alter our adherence to PLOS ONE policies on sharing data and materials.

Last revision

I will reply to each point as follows:

Journal Requirements:

I double-checked with the literature used in the first submission and could not find any difference in the cited literature. Please let us know if this is a general comment or whether you specifically refer to our reference list- and which specifically you think has been changed.

Reviewers' comments:

Reviewer's Responses to Questions

Comments to the Author

1. If the authors have adequately addressed your comments raised in a previous round of review and you feel that this manuscript is now acceptable for publication, you may indicate that here to bypass the “Comments to the Author” section, enter your conflict of interest statement in the “Confidential to Editor” section, and submit your "Accept" recommendation.

Reviewer #2: (No Response)

2. Is the manuscript technically sound, and do the data support the conclusions?

Reviewer #2: Yes

3. Has the statistical analysis been performed appropriately and rigorously? 

Reviewer #2: Yes

4. Have the authors made all data underlying the findings in their manuscript fully available?

Reviewer #2: Yes

5. Is the manuscript presented in an intelligible fashion and written in standard English?

Reviewer #2: Yes

6. Review Comments to the Author

Reviewer #2: Thanks for your revisions, which I feel have greatly improved this ms. I do think the term "transfer activity/route" either needs defining earlier on or substituting with "commuting" which I think is more commonly used.

Thank you for answering our question concerning this term. We now changed it to „commuting“ throughout the entire manuscript.

I did have a query about the comparison on the % of flights passing within 10 m of the turbine (ref 25). Using % as a metric is only a fair comparison if both recording set-ups have exactly the same field of view. In the methods you state that bats could be detected upto 20 m from the turbine. Was this the same with the study outlined in ref 25? Clearly, the narrower the field of view the the higher the proportion of flights will be recorded within 10m of the turbine.

Good point and I apologize for overlooking the query in the first revision. Although the technical equipment was identical in both studies, the field of view around the turbine would have been different simply due to the different heigth of the turbine (� the taller SWT turbine was a few meters further away from the camera simply because of the height, even though the aperture angle of the camera was identical). Thus the field around the heigher turbine in the reference was wider. As the field of views are different, percentage as a metrics is not a good comparison, i agree. We now remove the direct comparison of flight paths percentage/numbers in relation to turbine distance between both experiments and focus more on the overall difference in bat activity, exploratory behavior and rotor passes (L. 284-290).

7. PLOS authors have the option to publish the peer review history of their article (what does this mean?). If published, this will include your full peer review and any attached files.

Do you want your identity to be public for this peer review? For information about this choice, including consent withdrawal, please see our Privacy Policy.

Reviewer #2: No

Additonal changes

I detected a grammatical error in Line 294 and changed it (hashave).

---

## [Editor Report · Decision Letter 2]

14 Jun 2021

Collision risk of bats with small wind turbines: worst-case scenarios near roosts, commuting and hunting structures

PONE-D-20-40193R2

Dear Dr.  Stefanie Andrea Harmtman

Thank you for carefully attending to and addressing the reviewer's comments.   Nicely done.

Brock Fenton

We’re pleased to inform you that your manuscript has been judged scientifically suitable for publication and will be formally accepted for publication once it meets all outstanding technical requirements.

Kind regards,

Brock Fenton

Academic Editor

PLOS ONE
---

## [Editor Report · Acceptance letter]

17 Jun 2021

PONE-D-20-40193R2 

Collision risk of bats with small wind turbines: worst-case scenarios near roosts, commuting and hunting structures 

Dear Dr. Hartmann:

I'm pleased to inform you that your manuscript has been deemed suitable for publication in PLOS ONE. Congratulations! Your manuscript is now with our production department. 

Kind regards, 

on behalf of

Dr. Brock Fenton 

Academic Editor

PLOS ONE